# Strategies to address recruitment to a randomised trial of surgical and non-surgical treatment for cancer: results from a complex recruitment intervention within the Mesothelioma and Radical Surgery 2 (MARS 2) study

Nicola Mills [1], Nicola Farrar,[1] Barbara Warnes [1], Kate E Ashton [1],[1] Rosie Harris,[1] Chris A Rogers [1],[1] Eric Lim,[2,3] Daisy Elliott [1]

¹Bristol Medical School, University of Bristol Faculty of Health Sciences, Bristol, UK
²Academic Division of Thoracic Surgery, The Royal Brompton Hospital, London, UK
³National Heart and Lung Institute, Imperial College London Faculty of Medicine, London, UK

**Correspondence to**
Dr Nicola Mills;
nicola.mills@bristol.ac.uk

## ABSTRACT

**Objectives** Recruiting to randomised trials is often challenging particularly when the intervention arms are markedly different. The Mesothelioma and Radical Surgery 2 randomised controlled trial (RCT) compared standard chemotherapy with or without (extended) pleurectomy decortication surgery for malignant pleural mesothelioma. Anticipating recruitment difficulties, a QuinteT Recruitment Intervention was embedded in the main trial phase to unearth and address barriers. The trial achieved recruitment to target with a 4-month COVID-19 pandemic-related extension. This paper presents the key recruitment challenges, and the strategies delivered to optimise recruitment and informed consent.

**Design** A multifaceted, flexible, mixed-method approach to investigate recruitment obstacles drawing on data from staff/patient interviews, audio recorded study recruitment consultations and screening logs. Key findings were translated into strategies targeting identified issues. Data collection, analysis, feedback and strategy implementation continued cyclically throughout the recruitment period.

**Setting** Secondary thoracic cancer care.

**Results** Respiratory physicians, oncologists, surgeons and nursing specialists supported the trial, but recruitment challenges were evident. The study had to fit within a framework of a thoracic cancer service considered overstretched where patients encountered multiple healthcare professionals and treatment views, all of which challenged recruitment. Clinician treatment biases, shaped in part by the wider clinical and research context alongside experience, adversely impacted several aspects of the recruitment process by restricting referrals for study consideration, impacting eligibility decisions, affecting the neutrality in which the study and treatment was presented and shaping patient treatment expectations and preferences. Individual and group recruiter feedback and training raised awareness of key equipoise issues, offered support and shared good practice to safeguard informed consent and optimise recruitment.

**Conclusions** With bespoke support to overcome identified issues, recruitment to a challenging RCT of surgery versus

## STRENGTHS AND LIMITATIONS OF THIS STUDY

⇒ Embedding a complex recruitment intervention into a randomised trial deemed difficult to recruit to enabled key challenges to be identified and addressed in 'real time'.

⇒ Findings were triangulated from multiple qualitative and quantitative data sources.

⇒ Over one-third of health care professionals approached were not interviewed and we did not have audio recordings of consultation discussions from half of the study sites.

⇒ Ability to feedback and engage with individual recruiters on recruitment to study discussions was not always possible, resulting in a written report being sent with no confirmation it was read.

no surgery in a thoracic cancer setting with a complex recruitment pathway and multiple health professional involvement is possible.

**Trial registration number** ISRCTN ISRCTN44351742, Clinical Trials.gov NCT02040272.

## INTRODUCTION

Challenges of recruiting to randomised controlled trials (RCTs) are well recognised.[1–3] Just over a quarter of respiratory trials and half of surgical trials fail to recruit to their target sample.[4–7] Recruitment failure is reported as the prevalent reason for premature closure of surgical trials.[8] Challenges tend to be more pronounced in trials comparing markedly different interventions, such as surgical versus non-surgical treatment, where issues around equipoise are likely to be heightened.[9]

Malignant pleural mesothelioma is a rare cancer of the lining of the chest wall affecting over 2500 people each year in the

UK.[10] Prognosis is poor with long-term survival ranging from 4 to 18 months and a 5-year survival rate of less than 10%.[11 12] While there have been advances in systemic treatments, surgery in the form of pleurectomy decortication (removal of the diseased lining of the chest and lung) or extended pleurectomy decortication (involving additional removal of the lining of the heart and/or diaphragm), remains the most commonly performed procedure worldwide in an attempt to improve survival despite no randomised evidence of its effectiveness.

The Mesothelioma and Radical Surgery 2 (MARS 2) multicentre RCT (ISRCTN 44351742) was conducted in the UK to determine whether (extended) pleurectomy decortication alongside chemotherapy is superior to chemotherapy alone with respect to overall survival for patients with pleural mesothelioma.[13] The study anticipated recruitment challenges given the interventions being very different, so a QuinteT Recruitment Intervention (QRI)[14] was embedded in the main phase of the study to support recruitment. The QRI is a flexible, tailored complex intervention that triangulates multiple qualitative strategies and quantitative data to identify and address recruitment difficulties as they arise in real time.[14 15] Having been embedded in over 70 trials, it has led to insights into recruitment issues and the development of targeted strategies that have improved recruitment.[14 16] The aim of the QRI in MARS 2 was to understand the recruitment process and how it operates in clinical centres, so that sources of difficulties could be identified, and suggestions made to optimise the process. Despite anticipated and identified challenges with recruitment, the MARS 2 study successfully recruited to target with a 4-month COVID-19 pandemic-related extension. This paper illustrates the key challenges and describes the actions undertaken to mitigate barriers to support the conduct of this, and future trials, with divergent treatment arms in a cancer setting.

## METHODS
### The MARS 2 study
Recruitment pathway: The MARS 2 study has been detailed previously.[13] Figure 1 summarises the typical study recruitment pathway in the context of usual clinical practice. Adults with a diagnosis of malignant pleural mesothelioma were mostly introduced to the study by respiratory physicians and/or oncologists alongside local research staff at one of 25 medical sites across the UK. Patients were then referred to a thoracic surgeon at one of five trial accredited UK surgical sites (often in different hospitals to the medical site) to determine eligibility and discuss the study in more detail before typically being referred back to the local medical team for study consent. Following two cycles of chemotherapy, consented patients were reassessed for eligibility and randomised to continue with chemotherapy alone or receive surgery and further chemotherapy. The study opened to recruitment in May 2015 with an initial 2-year internal pilot

phase and continued through the main study phase until the recruitment target of 328 patients had been reached. The planned recruitment end date was 31 July 2020, but recruitment was paused March–June 2020 due to the COVID-19 pandemic. Once the pause was lifted, sites restarted recruitment in a staggered way, depending on capacity, and recruitment was extended until 31 December 2020. The recruitment target was achieved in November 2020.

Flow of participants to the point of randomisation: 1030 patients were assessed for eligibility, 645 were eligible and 335 were randomised. Of the 310 eligible patients who did not consent, most were not approached for consent (n=183), did not consent (n=74) or did not undergo two cycles of chemotherapy and repeat CT scan (n=36).

### Patient and public involvement
Patient and public representatives were involved in the design of the trial, selection of the primary outcome measure and the definition of the minimally important difference in relative survival, and development of patient facing study documents. The trial steering committee also included a patient and public involvement representative.

### The QRI in the MARS 2 study
The QRI in MARS 2 was only integrated within the main phase of the trial (not pilot) and entailed two core phases as detailed elsewhere.[13 14] QRI phase 1 aimed to understand the recruitment process at sites and key challenges that had the potential to hinder recruitment. Methods included mapping eligibility and recruitment pathways, interviewing a purposive maximum variation sample of study/site staff and patients, audio recording study recruitment consultations and reviewing patient-facing documentation. Topic guides, adapted from previous QRI studies and refined to explore emerging findings as the study progressed, were used flexibly in the interviews. Interview and recruitment consultation recordings were transcribed verbatim, and along with recruitment screening logs, were subject to simple counts, content and thematic analyses led by NM.[15 17 18] Particular attention was paid to the consultation recordings for instances of unclear, insufficient or imbalanced information provision and transferring of clinician treatment biases to patients. Preliminary analysis, drawing together data from the various sources to identify and understand common challenges, informed further data collection. NM, NF and DE, as experienced qualitative/QRI researchers who conducted the interviews, independently analysed a proportion of transcripts to assess the dependability of coding, and met regularly to review coding and descriptive findings, agree on further sampling and discuss theoretical development alongside findings from screening logs.

QRI phase 1 commenced in March 2018. Findings were fed back to the chief investigator (CI) and trial management group (TMG) as key issues arose. Effective strategies, tailored to address identified issues, were devised

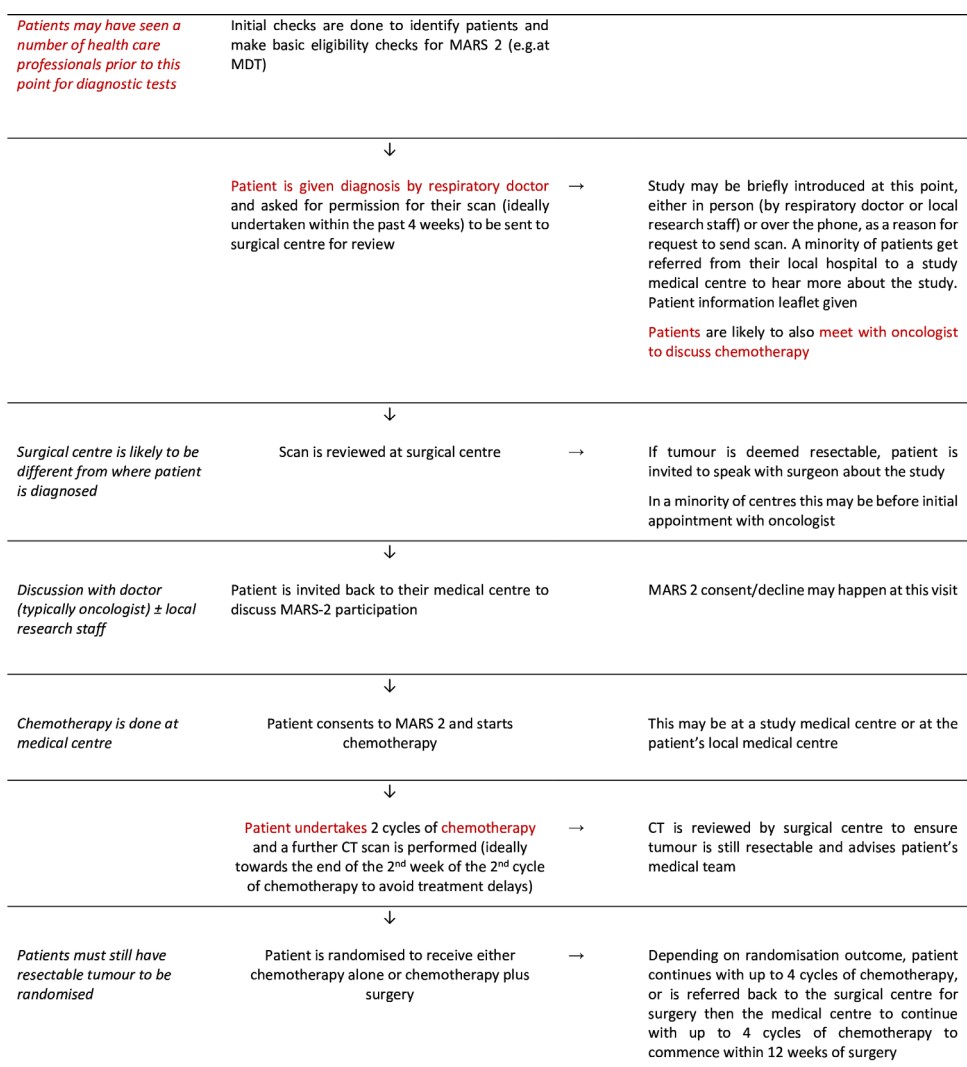

**Figure 1** MARS 2 recruitment pathway (with typical clinical pathway in red). MARS 2, Mesothelioma and Radical Surgery 2; MDT, multidisciplinary team.

and implemented from October 2018 (start of QRI phase 2). Phases 1 and 2 continued cyclically until recruitment target was reached. QRI training, based on the QuinteT RCT recruitment training intervention,[19] was additionally delivered prior to QRI phase 1 (November 2017 to February 2018) to tackle barriers that had emerged in the pilot phase[20] and previous QRI studies.[21–25]

## RESULTS
### QRI sample
As part of the QRI, 19/25 study sites consented to audio record recruitment discussions. We obtained 55 consultation recordings with 16 study recruiters from 12 sites lasting a mean of 49 min. Between December 2017 and March 2020, we undertook 25 interviews with 24 study staff (of 39 invited) from 18 sites lasting 40 min on average. The sample consisted of nine oncologists, six research nurses/practitioners/co-ordinators, five surgeons, three respiratory/chest physicians and one TMG member. Additionally, we conducted 21 interviews with 20 patients

from 5 sites who had been invited to participate in MARS 2. Findings from patient interviews have been reported elsewhere,[26] although they fed into the interpretation of findings presented herein.

### Recruitment challenges
MARS 2 was described by health professionals as an important study that answers a much-needed question as to the role of pleurectomy decortication for mesothelioma with the potential to change future practice. It was recognised that surgery was currently undertaken in the UK in an 'ad hoc' and 'unregulated way' that was unsupported by high-quality evidence. Building the evidence base to address this was the main drive to be part of the study. Challenges with recruitment, however, were evident. These were apparent at all stages of the recruitment process from patient identification to receiving consent and were due largely to intellectual and emotional challenges around equipoise compounded by a complex pathway that entailed patient contact with

different health professionals and conflicting treatment views.

## Organisational challenges of conducting mesothelioma research
### Complex pathways
The MARS 2 recruitment pathway involved additional steps to usual clinical practice (figure 1). Many site staff described recruitment challenges stemming from the complex study pathway that involved receiving and referring patients for time-sensitive investigations and treatment from different specialists and hospitals. Some patients were referred to MARS 2 sites by local centres which were not involved in the RCT, meaning their standard of care was arranged through other hospitals with additional visits related to MARS 2 being carried out at study sites. Study sites relied on colleagues from neighbouring hospitals being aware of MARS 2 and referring patients for consideration. Established regional mesothelioma multidisciplinary team (MDT) meetings were recognised as crucial for identifying potentially eligible patients, but where sites were not operating within this structure referrals were sometimes more ad hoc leading to missed patients or delayed referrals making patients ineligible (table 1, quote 1). Pressures of a busy service were noted as exacerbating the problem. Trying to fit the MARS 2 study and its complex pathway and timings into a service that was perceived as 'already stretched' was deemed challenging. Delays in investigations and chemotherapy meant that some patients became ineligible, or they declined the study as they were not willing to accept further treatment delays. Staff capacity issues resulted in one site having to temporarily pause receiving referrals for study consideration. As recruitment progressed, investigator fatigue was offered as an explanation for a slowdown of recruitment and a potential for 'coasting', if sites met what they set as their recruitment target.

### Competing research agendas
With the 'explosion of interest in immunotherapies' and molecular treatment showing promise, a competing research agenda was proposed as hindering recruitment. Studies that should have been complementary and not overlapping in terms of patient selection were considered by some as competing, with a minority of clinicians stating that they either decided which study the patient would be best suited for or they presented the options for the patient to decide.

## Tension between clinical versus personal equipoise
### Hesitancy in referring patients for study consideration
Doubts about the effectiveness of surgery for mesothelioma were evident across specialties. These were driven by findings from previous RCTs showing other types of surgery as not improving survival[27 28] and potentially causing harm,[28] and a belief that the condition should not be treated with chemotherapy or surgery due to its poor prognosis. The impact of these views was recognised at all stages of the recruitment process. They were felt

to account for fewer than expected study referrals from neighbouring sites and colleagues, or patients coming to discuss the study with a firm idea of their treatment plan, which often accorded with the previous specialist's view (table 1, quotes 2, 3). This was described as making the study discussion challenging.

### Bias in determining eligibility
Although study recruiters described uncertainty around the effectiveness of pleurectomy decortication across the clinical community (ie, clinical equipoise), and therefore, a need for the study, individual levels of equipoise varied. These ranged on a continuum from not believing the surgery will be effective, to being neutral as to its effectiveness, having a 'hunch' that it will be effective (especially regarding a subset of 'young and fit' patients), and in a minority of cases, having a seemingly strong belief in the effectiveness of the surgery. Personal levels of equipoise, though, could vary depending on the individual patient. This created a tension at times between appreciating the clinical equipoise versus personal feelings about what would be right for a particular patient. Recruiters shared their discomforts in deeming a patient eligible when it went against their experience and 'gut feeling' about what they felt would be right for that individual. Older patients, those less physically fit and those with a sarcomatoid cell-type caused the most discomfort and were less likely to be put forward for the study, even though 'on paper' they fulfilled the eligibility criteria (table 1, quotes 4, 5).

### Recruiter bias in describing treatments
Bias was also evident at times in the description of treatments given in patient consultations. Study recruiters recognised that they may not have always been in equipoise about surgery but were keen to set any biases aside and explain the study impartially. In practice, although they articulated equipoise to patients by explaining the uncertainty as to the effectiveness of lung-sparing surgery, the equipoise was sometimes compromised in the ensuing discussion. This was often indirectly, through choice of words that created an imbalanced description of treatments, for example, referring to surgery as 'giving an extra benefit' as opposed to having 'what traditionally is given for this and hoping for the best' or describing it as 'experimental' and a 'pretty horrible operation'. In a very few cases equipoise was over-ridden in the study discussion, with the recruiter making clear their treatment biases and steering patients in a particular direction that accorded with their beliefs for that individual. Unsurprisingly, patients tended to go in the direction of the clinician's steer. A minority of clinicians also offered patients surgery off trial, bringing into question their level of equipoise (table 1, quotes 6, 7).

### Discomfort approaching patients at time of diagnosis
Finally, there was an indication that some recruiters, more so research nurses/co-ordinators, were uncomfortable

**Table 1** Quotes from interviews supporting identified recruitment challenges

| Quote | Organisational challenges of conducting mesothelioma research |
|---|---|
| 1 | 'I actually came across some patients who had been referred too late and that's because they've been, in absolutely good faith from our point of view, mismanaged in small peripheral district hospitals. I know there has been a lot of effort put into getting them onboard… When you get a patient that's been moved back and forward from district general hospital for about six months, at that point he's either progressed too much or simply deteriorated too much not to be able to have any treatment at all. That is a failure, not just of the trial' (Surg06 at interview) |
| | **Equipoise-related issues** |
| | *Referral hesitancy* |
| 2 | 'One thing I've been slightly disappointed about is the lack of referrals from my neighbouring Trusts, because I had expected more….I think there is some reluctance from the oncologists in some of the neighbouring Trusts about the MARS 2 study… I think that potentially they don't believe in surgery….I guess probably because of a lack of evidence of any good from it and there is some evidence of harm' (Onc04 at interview) |
| 3 | 'The other patients are coming from local hospitals and have been discussing the mesothelioma with some other chest physicians. Most of the time, they are already coming with the idea of not wanting anything done. That is a really difficult conversation in the clinic, because clearly the patient already has his own idea of, 'The other doctor told me that I'm going to die. I want to just live the rest of my life in the best way possible. I don't really want to discuss too much with you" (Surg04 at interview) |
| | *Discomforts around eligibility criteria* |
| 4 | 'I do get patients I see who are eligible and I look at them and think, 'No, you're not going to get through this easily.' …Sometimes you do suggest to patients, 'Although on paper you're okay, I don't think you're fit. You meet the criteria for eligibility, but I just don't think you're strong enough" (Surg01 at interview) |
| 5 | 'Sending patients [with sarcomatoid cell type] to [surgical site] away from their families for something that I can't guarantee will do them any good when I know their prognosis is rubbish whatever we do to them, doesn't seem very sensible to me' (Onc04 at interview) |
| | *Imbalanced description of treatment options* |
| 6 | PATIENT: And I was thinking about it and I talked it over with the lads, and I thought, you know, I think I'd just rather go for the chemotherapy<br>ONCOLOGIST 25: Yeah. I don't think that's the wrong thing for you, to be honest…. I would be more than happy to offer you chemotherapy treatment. I think adding surgery into the mix, particularly when it's quite a bit further away, is probably an additional complication. It's maybe not quite the right thing for you. And it sounds like you've come to that conclusion yourself. And I think that's the right thing.<br>ONCOLOGIST 25 to researcher after: 'My patter was rather deliberately weighted a bit on the negative side [in describing surgery] as I really didn't think this trial was the right thing for this patient. So rightly or wrongly the pitch does go along those lines as in practice the surgeons tend to take them on' |
| 7 | WIFE: [Surgeon] didn't think it was fair to put him in the trial because he would only get a 50% chance of that operation, and he thought he had a 75%, probably 85% or whatever, better chance of survival or prolonging his life with that operation. So it was [surgeon] that decided not to put him in the trial….<br>PATIENT: His words were, 'Mr 'Baldwin', I just put a few years on your life |
| | **Discomfort approaching patients** |
| 8 | 'There's a lot going on, they're newly diagnosed patients, we've hit them with a massive amount of information, a massive amount of life-changing information… So if they're feeling a bit like they've got information overload already, then pushing them down that route, you don't want to make them more distressed. Sometimes it's not right for everyone…. I have had one or two patients where we've got a little way into the process and I've got a feel that they're- and I will say to them, 'This is not compulsory. If you feel that it's all a bit much for you and it's a bit overwhelming then maybe the trial's not for you, and that's fine. You just need to let me know that you don't want to take it any further'. That has happened with one patient and he said, 'Do you know what? You're right. I really don't want to think about it right now'. And that's fine…. What I do then is just pop them back into the standard of care system' (RN/P/C13 at interview) |

approaching patients about the study at the time of receiving a life-limiting diagnosis. If they sensed the patient was overwhelmed, then the conversation about MARS 2 (if raised) was sometimes couched in such a way that recognised this and offered patients a clear way out (table 1, quote 8).

### Addressing identified recruitment challenges

The recruitment issues identified during phase 1 of the QRI were fed back to the CI and TMG from September 2018 onwards on a regular basis and actions agreed to address them (phase 2). Table 2 presents an outline and timelines of the main QRI-informed actions that

**Table 2**  Key QRI-informed actions with timelines in MARS 2

| Date | Action | Delivered to (mode) | |
|---|---|---|---|
| October 2017 | Feedback on patient information leaflet to ensure non-leading descriptions of study treatments | Trial team (written) | QRI-informed actions prior to/during phase 1 |
| November 2017 | Recruitment session to raise awareness of common RCT recruitment challenges and tips to deal with them at MARS 2 investigator's meeting | Recruiting site staff (face to face) | |
| January 2018 | 1-day QRI-informed trial recruiter training day open to any surgeon who recruits to RCTs (not specific to MARS2) | 3 recruiters from 2 sites (face to face) | |
| February–September 2018 | Site initiation or refresher visits with short presentation and discussion to raise awareness of likely recruitment challenges and QRI-informed tips to deal with them | 9 sites (7 video; 2 face to face) | |
| April 2018 | Tips document for discussing the study with patients based on anticipated issues | Recruiting site staff (email) | |
| September 2018 | Suggested changes to text for CRUK (Cancer Research UK) webpage to ensure clear and balanced study description | Patients (webpage) | |
| October 2018–October 2019 | Monthly recruitment tips in study newsletter addressing issues raised from QRI Phase 1 | Recruiting site staff (email) | QRI-informed actions in phase 2 |
| October 2018–January 2020 | Individual study recruitment consultation feedback | 11 recruiting staff from 13 sites (email, 1 verbal) | |
| October 2018 | Site feedback of equipoise issues from study recruitment consultation recordings | All recruiting staff at 1 site (written, verbal) | |
| November 2018 | Phase 1 feedback and discussion on equipoise at surgeons meeting | 3 MARS 2 surgeons of 12 invited (face to face) | |
| January 2019 | Phase 1 feedback and discussion on key identified issues at BTOG (British Thoracic Oncology Group) conference meeting | 12 attendees (face to face) | |
| March 2019 | Recruiting tips document updated to address key findings from phase 1 | Recruiting site staff (email) | |
| May 2019 | Phase 1 feedback and discussion at investigator's meeting | Recruiting site staff (online) | |
| June–October 2019 | Site feedback visit with a focus on equipoise issues | 4 sites (face to face) | |
| July 2019 | Feedback and discussion of patient interview findings | Nurse/co-ordinator recruiting staff (online) | |
| September 2019 | Sharing best practice in response to common patient questions at recruitment | Recruiting site staff (email) | |
| November 2019 – September 2020 | Visually enhanced monthly newsletters with more emphasis on QRI findings and extracts of good recruitment discussions | Recruiting site staff (email) | |
| February 2020 | Personalised motivational emails to sites noting areas of exceptional recruitment practice and offering suggestions on areas to focus on | Recruiting site staff (email) | |
| February 2020 | Targeted emails from the CI to address issues identified from ongoing QRI interviews/discussions with sites | Recruiting site staff (email) | |

CI, chief investigator; MARS 2, Mesothelioma and Radical Surgery 2; QRI, QuinteT Recruitment Intervention; RCT, randomised controlled trial.

were implemented. A mix of face-to-face, online video, telephone and written documentation was used to feedback on findings and engage with study team members at individual, site and whole study levels to discuss and address issues. Support focused on maximising the pool of eligible patients in terms of finding, assessing and approaching them by offering practical solutions, for example, sharing site-initiated ideas such as viewing and flagging potentially eligible patients from MDT lists in advance and regular contact with lung nurses to ensure patients were not missed from being assessed for eligibility. Or by raising awareness using evidence from QRI

data and opening discussion of how biases and discomforts affected whether patients were considered eligible for the study and approached about it. Awareness was also raised of patients being primed towards or against treatment from colleagues prior to study consideration to indicate the value of recruiters exploring what patients had already been told about treatment options, and to promote sharing of information on MARS 2 with wider colleagues especially those encountered by the patient earlier in the pathway.

Support also focused on ensuring that study discussions offered full, clear and balanced information. There was a particular emphasis on issues around equipoise and good practice for conveying it. In both individual and group feedback sessions, QRI anonymised data were presented to raise awareness of how imbalances of treatment descriptions, for example, disproportionate emphasis on potential risks and benefits or language that subtly favoured or deterred a treatment, could steer patients away from the study and towards or against a particular treatment. Examples from consultation recordings were shared and discussed of more neutral ways of describing treatments and of engaging with patient treatment preferences. The recruitment tips document (online supplemental file 1) offered suggestions for structuring and discussing recruitment consultations in a way that emphasised the uncertainties around the best treatment, with specific aspects of this spotlighted in monthly newsletters using extracts of QRI data. Feedback and support were set within the recognition of site teams and continued above target achievement and encouragement to keep on track.

After the COVID-19 study pause was lifted, actions focused on positive encouragement and motivation to help raise the profile of MARS 2 for the final stages of recruitment, respecting the pandemic-related pressures that sites were under and the varying capacity that they subsequently had.

## Impact of QRI on optimising recruitment and informed consent

Figure 2 shows actual recruitment against target, depicting points of QRI-informed actions that were delivered at greater than an individual recruiter and single site level. Actions that were developed and delivered in response to findings from QRI phase 1 began in October 2018 (start of QRI phase 2). During QRI phase 2, there were eight QRI-informed actions delivered beyond individual and single site level, as well as monthly recruitment tips in newsletters. There were also a further 22 actions delivered at individual recruiter and site level (largely tailored feedback of screening log and/or consultation recordings) not depicted on this graph (see table 2).

The QRI was integrated throughout the main phase of MARS 2 (including set up) with several actions delivered at study-wide level, so it was difficult to evaluate the impact it had on recruitment. The timing of QRI actions in relation to the recruitment rate in figure 2 suggests that the actions may have contributed to increasing and maintaining recruitment above target prior to the pandemic. From screening log data, the average number of patients recruited per month per site was 0.27 in September 2018 (just prior to QRI phase 2 and the roll out of feedback and training), which rose to 0.65 in March 2020 (after the delivery of most of the QRI feedback and training and just before the study pause). Analysis of recruitment figures 6 months before and after recruiters received individual feedback on their study recruitment consultations suggest a dose-response effect. Numbers randomised before and after feedback were not dissimilar in the five sites where only one recruiter had received individual

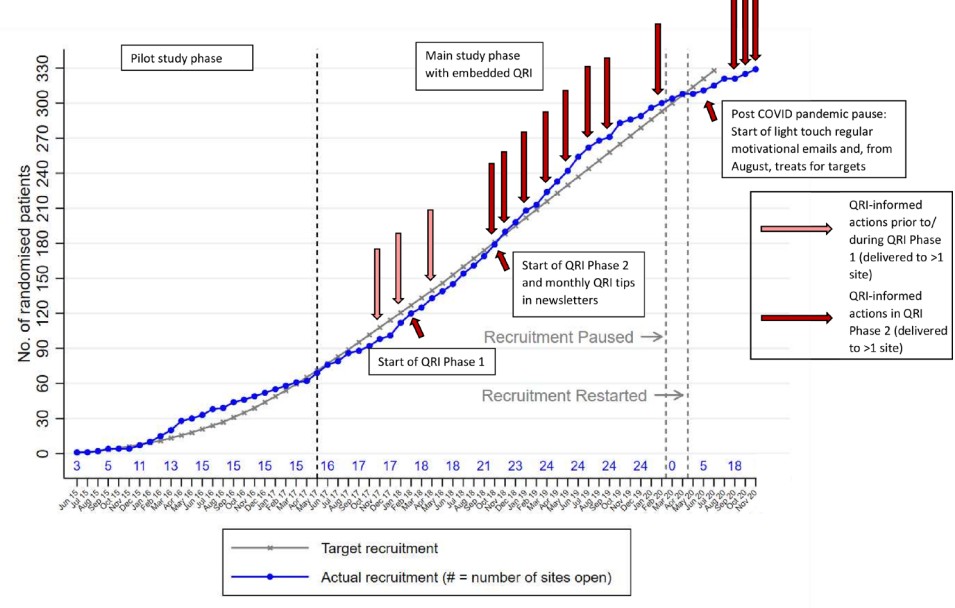

**Figure 2** Recruitment to MARS 2 study against recruitment target depicting points of QRI-informed actions beyond individual and single site level. MARS 2, Mesothelioma and Radical Surgery 2; QRI, QuinteT Recruitment Intervention.

feedback at the time of analysis. The greatest difference was observed in the one site where feedback was given at the same time point to all three recruiters (both individually and as a group). At this site, no patients were randomised in the 17 months before feedback and six patients were randomised in the 5 months after, which represented 43% of their total recruitment over 5.5 years.

Several recruiters who received feedback on their individual recruitment discussions acknowledged the value of it, stating that it had raised awareness of certain issues and that they intended to alter their practice accordingly. An evaluation of before-after tailored feedback showed evidence of advice being implemented and improvements in terms of how the study was introduced, balancing of treatment descriptions, explanation of randomisation and avoidance of potentially problematic terminology (table 3). We also identified suggestions and strategies that were given within wider recruitment training sessions and recruitment tips/guidance documents being implemented in study recruitment discussions.

For a very small number of sites (n=4), the recruitment challenges were elicited and discussed but were considered insurmountable. They related to a combination of political, logistical or personal factors that were beyond the scope of the QRI. This resulted in two sites closing to recruitment early, and two sites not recruiting any patients.

## DISCUSSION

This study offers important insights into the complexities of conducting research and how, with bespoke support to overcome identified issues, recruitment to a challenging RCT of surgery vs no surgery in a thoracic cancer setting is possible. The MARS 2 study had to fit within the framework of a service that was considered overstretched where patients encountered multiple healthcare specialties and treatment views along their diagnostic and treatment pathway, all of which challenged recruitment. With advances in medical therapies, the study had to compete with other research studies, draining already tight resources and reducing the pool of potential participants. Research staff had to be comfortable approaching patients about a study at a time when their survival was limited and discussing treatment that they may not have felt was appropriate for all. Specific to MARS 2 was the dependence on time-sensitive tests and investigations, and involvement of surgical specialists often at different hospitals, which further added to the complexities of conducting the research. Having identified key context and trial-specific issues, QRI-informed actions were devised and implemented to raise awareness, share good practice and offer support in a bid to safeguard informed consent and optimise recruitment. The trial achieved recruitment to target with a 4-month COVID-19 pandemic-related extension.

One of the key findings was the varying levels of individual equipoise around the surgical intervention and the impact this had on study recruitment. Despite describing treatment uncertainty within the clinical community, it was clear that not all those involved with the care of mesothelioma patients fully subscribed to this view at a personal level, or in relation to particular patients. In the absence of robust and reliable evidence for pleurectomy decortication, individual levels of equipoise were often shaped by past clinical experiences and findings from earlier trials evaluating other types of surgery for mesothelioma which did not infer survival benefits[27 28] and indicated possible patient harm.[28] Some appeared to struggle with referring, assessing and discussing the study with patients when it conflicted with their clinical judgement. Consequently, there were patients fulfilling the eligibility criteria who did not have the opportunity to consider participating in the study and others who did not receive the full and balanced information necessary for informed decision-making. Role conflict and challenges with setting aside personal views when determining study eligibility and conveying balance during study and treatment discussion have been identified in other RCTs set within the thoracic and broader cancer/surgical contexts and noted to adversely impact recruitment.[22 23 29–34] These issues are heightened in trials comparing very different interventions.[9] In such trials (including MARS 2), clinicians and patients are more likely to struggle with achieving a position of equipoise because of factors such as strong clinical specialty convictions and a priori treatment preferences which in turn may influence how information about the trial is portrayed and patients' decision to participate.[9]

Introducing a randomised trial to a population of mostly older patients who have been subjected to a multitude of investigations from different clinical specialties and diagnosed with a poor-prognosis cancer can be challenging. Interviews with patients conducted in the main phase of MARS 2[26] concurred with findings from the pilot phase showing that participants had difficulties with the volume and complexity of study information, including understanding of equipoise and randomisation.[20] Audio recordings of recruitment discussions as part of the QRI in the main trial phase offered insight into these findings, demonstrating at times biased descriptions of treatment options and unclear explanations of randomisation which would have contributed to the observed patient difficulties.

MARS 2 recruitment was further challenged by the condition under investigation being a rare cancer. The rarity of the condition makes finding and recruiting enough patients difficult. Rare disease trials tend to be smaller, longer and more prone to being terminated, withdrawn or suspended compared with those for non-rare conditions.[35] Missed opportunities for referring and approaching patients, as identified in this study, become even more impactful when the pool of potentially eligible patients is limited at outset. To add to this challenge, the MARS 2 study investigated a condition that was very active in terms of research to advance medical treatment opportunities. We know from clinician interviews and

**Table 3** Implementation of QRI-informed suggestions in recruitment consultations

| Before individualised feedback | After individualised feedback |
|---|---|
| **Introducing the study and rationale** | |
| *Now with the trial, which is called MARS-2, part of the trial is that you have the chemotherapy straight away, first half of the chemotherapy, then we randomise you either to surgery or not surgery. If you go for surgery have that and then you have further chemotherapy after. If you get the arm of the trial where there is no surgery you just complete the chemotherapy with [oncologist] which is all given here. Surgery, we would have to ask our surgical colleagues who are based in [city] whether they think looking at your scan surgery is an option. And if they did then we would put you in to the trial and at that point of randomisation make a decision as to whether you go in for surgery or not (Resp02)* | *And the third option is a trial that we are taking part in, which looks at the combination of surgery with chemotherapy vs chemotherapy…So the process for the…I meant to say study…for the study is to compare the standard treatment which is chemotherapy and then you get randomised either to chemotherapy with standard treatment, or chemotherapy, surgery and then the rest of the chemotherapy. All the chemotherapy would be here, the surgery would be done at [city]….So, they look to take away the mesothelioma and the lining of the lung on that side. It is done with the hope to cure it, again, it is about trying to take away as much as possible…And the reason why it is a study is because we are not certain yet whether that adds anything or not. That is an unknown question (Resp02)* |
| **Balancing treatment descriptions** | |
| *I know that there's not many surgeons in the country that like doing this… it's a big, long operation. It's rare for it to be less than 5 or 6 hours. It can frequently be 7, 8, 9 hours long. It takes a lot out of [the surgeon]e as much as the patient. And, you know, it isn't to be sniffed at. There are risks. There are risks of infection in the space around the lung, in the wound, in the lung, pneumonia. There's risks of bleeding, we have to leave drains in to get the lung to expand. It often takes 3 drains for a couple of weeks before the bubbling stops. So it is a big operation. I liken it, for most patients, is - I'm afraid I'm brutally honest. It's like being hit by a bus. It is not a small operation. And it's not something that, you know, is clearly the best thing to do (Surg01)* | *We set off with the expectation that we can remove everything that we can see and feel, and hope that that is the case, but the question then is whether this is worth it… what are the risks and benefits of this operation? The risks are that it is a big operation…It does knock the stuffing out of patients….There are risks of infection, bleeding, pain, bubbling from chest drains. It's a huge operation. Therefore, we have to justify those risks in terms of benefits. The [possible] benefits are quality of life and length of life (Surg01)* |
| **Explaining randomisation** | |
| *If you go in the study you sign a consent, and if you move on to the second part, after the two cycles you're given a number and that number is put in to a computer that just randomly decides (Onc03)* | *Even when you think a treatment sounds logical and sensible, and a good idea, it's very important to assess it in a clinical trial because that's the only way you can really say whether a new treatment is better than the old, standard treatment, okay. So, to assess that, you have to make a comparison and in this study [explains study process]…and then you're randomly allocated to one of two groups. …. everyone gets the same amount of chemotherapy, but one group gets this surgery in addition, and at the end of the study you should be able to compare the two groups, and any difference between them should be down to the surgery (Onc03).* |
| **Avoiding potentially problematic terminology** | |
| *So, what the trial is looking at is, it's not a <u>trial</u> of whether the chemotherapy's useful or not in this situation. We're looking at whether surgery is an option in the future for patients. So basically, that's the, for want of another word, <u>experimental</u> part of the trial. Okay. So, the chemotherapy that is involved in the trial is actually standard, sort of, <u>standard treatment</u>…. Okay. And that's tried and tested chemotherapy. So, there's nothing new about that. That's our current treatment (ResStaff01)* | *So, obviously, Dr [respiratory physician] went through what your options are, and one of them was this study, the MARS 2 <u>study</u>. As she mentioned, if you go on to the study you will be allocated to one of two groups, one group will have some chemo and then go on to surgery, the other group would have some chemo then continue with the chemo (ResStaff01)* |

MARS-2, Mesothelioma and Radical Surgery 2; QRI, QuinteT Recruitment Intervention.

screening log data that a very small number of patients being considered for MARS 2 were lost to other studies, but we do not know how many patients were lost before MARS 2 was even considered. This puts a further strain

on recruiting for a trial that investigates a rare and life-limiting cancer where patients are already difficult to find and where there may be discomfort in approaching them about research at such a time. The value of employing

methods to understand and address recruitment challenges in such trials, as with the QRI in this study, is heightened, as are widespread campaigns to encourage patients to proactively seek involvement in clinical studies they might be eligible for.[36]

A core strength of the study was the ability to analyse actual practice (audio recording of recruitment discussions), as opposed to relying on reported practice, and to triangulate findings from multiple data sources to gain an in-depth and rich understanding of the key (and often hidden emotive[21 22]) difficulties. This was done in a relatively short time to enable strategies to be devised and implemented to support recruiting staff while recruitment was underway. The MARS 2 study took 5 years to randomise 335 patients. This is a threefold increase in the rate of recruitment compared with the next largest surgical trial for mesothelioma—MARS—which randomised 50 patients in 3 years.[28] The timing of QRI-informed actions against an increased and sustained recruitment rate is suggestive of a positive impact but we cannot rule out other factors that may have contributed to this (particularly the opening of new sites) or infer which interventions may have been more or less effective. Nor can we say if the recruitment target would have been reached without the QRI. The internal pilot phase did not integrate QRI and achieved target, although once the number of opened sites peaked recruitment started to steadily decline. Conversely, once the number of opened sites peaked in the main study phase, recruitment continued to steadily rise in time with the delivery of the QRI-informed actions. We also identified evidence of improved study information provision with clearer presentations of equipoise following individual or group-tailored consultation feedback, which is likely to have enabled better informed decision-making which may or may not translate into increased recruitment. We appreciate that over a third of the staff approached were not interviewed and we did not have audio recordings of consultation discussions from half of the sites. There may, therefore, have been further recruitment issues unique to these sites that we were unable to elucidate and address. Furthermore, we experienced difficulties in requesting to meet several recruiters to share individual consultation feedback, resulting in a written report often being sent with no confirmation it was read. Future research should focus on distilling the active components of complex recruitment interventions, defining how such interventions should be deemed successful (eg, increase in rate of recruitment or a measure of informed consent) and determining the best methods to evaluate their effectiveness.

One of the key findings from the QRI in MARS 2 was the impact that the wider clinical and research context had on views of the study and recruitment to it. Surgery for mesothelioma is controversial.[37] Findings from previous trials—showing other types of surgery for mesothelioma as not being beneficial with potential for harm[27 28]—contributed to the views held by some clinicians within and outside of the study on the appropriateness of surgery, and therefore, appropriateness of the study. Moreover, the study was set within a clinical context for a rare, life-limiting condition that had a convoluted pathway involving cross-specialty personnel and exposure to varying views on treatment. With promising new treatments on the horizon, the study also had to compete with other research studies within a resource-stretched service. Based on these findings, we recommend understanding the wider context of research at an early stage, both in terms of the clinical setting, pathway and prior research, and the potential impact that this could have on the success of a trial. Pretrial or early trial meetings and training workshops with site staff to detail current evidence and trial rationale, ascertain usual practice and engage with treatment views could help illuminate and mitigate potential difficulties at outset. Continued exploration along with recruiter feedback and training as recruitment is underway has the potential to help address ongoing discomforts. We also recommend engaging health professionals upstream of the study recruitment discussions in such activities, recognising the impact they can have on numbers of patients put forward for study consideration and patient treatment expectations.[38 39] Outside of a specific trial, broad RCT recruitment training raising awareness of common hidden challenges and strategies to overcome can offer additional opportunities to improve recruitment in future RCTs where recruitment is deemed to be challenging.[24 40]

## CONCLUSIONS

A complex multimethod intervention to optimise recruitment revealed important insights into the challenges of conducting a surgical randomised trial within the thoracic cancer setting. Recruitment occurred in what was considered a resource strapped, research competitive and logistically complicated clinical context. The QRI provided important insights into how clinician treatment biases, which were shaped by the wider clinical and research context alongside experience, had a noticeable adverse impact on multiple aspects of the recruitment process prior to and during study participation discussions. We were able to raise awareness of identified issues and support clinicians through feedback and training to overcome challenges for effective recruitment and informed decision-making. Recruitment to an RCT with very different treatment options set within a complex recruitment pathway with multiple health professional involvement is possible with bespoke training and support to address key equipoise issues.

**Acknowledgements** The MARS 2 study is sponsored by The Royal Brompton and Harefield NHS Foundation Trust. We would like to thank the clinical/research teams and patients at each hospital for their support with the QRI component of MARS 2.

We would also like to thank Professor Jenny Donovan for comments on an earlier draft of the manuscript.

**Contributors** NM, NF and DE are members of the University of Bristol QuinteT (Qualitative research integrated within Trials) research group and formed the QRI team within MARS 2. NM and DE led the design and conduct of the QRI. NF led the patient interview aspect of the QRI as part of her doctoral research. NM conducted the QRI, including data acquisition, analysis and interpretation, in discussion with DE and NF and drafted the manuscript. EL is the chief investigator for MARS 2, KEA and BW are the former/present trial managers, RH is the statistician and CAR is the methodological lead. All authors contributed to the design of MARS 2, including the QRI component, and drafting of manuscript for intellectual input. NM is the guarantor for the QRI research within the MARS 2 RCT,

**Funding** The MARS 2 study, with integrated QRI in the main study phase, was funded by the National Institute for Health and Care Research (NIHR) Health Techonology Assessment (HTA) programme (project number 15/188/31). The main phase of the MARS 2 study was designed and delivered in collaboration with the former Bristol Trials Centre Clinical Trials and Evaluation Unit (CTEU) (now the Bristol Trials Centre—BTC), a UK Clinical Research Collaboration (UKCRC) registered clinical trials unit which was in receipt of NIHR Clinical Trials Unit (CTU) support funding. DE was supported by the NIHR Bristol Biomedical Research Centre part way through. NF's PhD was funded by the Medical Research Council (MRC) Network of Hubs for Trials Methodology Research (HTMR) (MR/L004933/2/R9).

**Disclaimer** The views expressed are those of the authors and not necessarily those of the NIHR or the Department of Health and Social Care. The funder and sponsor had no direct involvement in study design; collection, management, analysis and interpretation of data; writing of the report; and the decision to submit this report for publication.

**Competing interests** None declared.

**Patient and public involvement** Patients and/or the public were involved in the design, or conduct, or reporting, or dissemination plans of this research. Refer to the Methods section for further details.

**Patient consent for publication** Not applicable.

**Ethics approval** This study involves human participants and research ethics approval for the MARS2 study which included the integrated QRI was granted by London—Camberwell St. Giles Research Ethics Committee (reference 13/LO/1481) on 7 November 2013. Participants gave informed consent to participate in the study before taking part.

**Provenance and peer review** Not commissioned; externally peer reviewed.

**Data availability statement** Data are available on reasonable request. Data, in terms of deidentified consultation and interview transcripts, are available from the corresponding author on reasonable request. This is on the condition that the request fulfils the necessary approvals in place for 'controlled access' data, that participants have agreed to the optional consent to share their anonymised data, and that participant anonymity or privacy is considered not to be compromised.

ORCID iDs
Nicola Mills http://orcid.org/0000-0002-2960-2940
Barbara Warnes http://orcid.org/0000-0002-1326-0448
Kate E Ashton http://orcid.org/0000-0002-9163-0512
Chris A Rogers http://orcid.org/0000-0002-9624-2615
Daisy Elliott http://orcid.org/0000-0001-8143-9549

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
