## [Reviewer comments · BMJ Open]

ARTICLE DETAILS

TITLE (PROVISIONAL)	Strategies to address recruitment to a randomised trial of surgical and non-surgical treatment for cancer – results from a complex recruitment intervention within the Mesothelioma and Radical Surgery 2 (MARS 2) study
AUTHORS	Mills, Nicola; Farrar, Nicola; Warnes, Barbara; Ashton, Kate; Harris, Rosie; Rogers, Chris; Lim, Eric; Elliott, Daisy

VERSION 1 – REVIEW

REVIEWER	Yip, Kay Por University of Birmingham, Institute of Inflammation and Ageing
REVIEW RETURNED	09-Oct-2023

GENERAL COMMENTS	The authors describe the challenges uncovered in a mixed method approach to investigate recruitment obstacles in the MARS 2 randomised controlled trial. The manuscript highlights how QRI has improved recruitment rate in the trial that it reaches target recruitment numbers. It also describes nicely how unconscious biases and underlying views of clinicians and trial staff can adversely impact recruitment, particularly in the role of cancer trials. I recommend that this manuscript should be accepted. Below are a small list of recommendations proposed 1) First point in the 'Strengths and Limitations' of the abstract will need to be reworded as it is difficult to follow2) Page 6 line 34 - Findings from Phase 1 were and fed back to the Chief Investigator (CI) and trial management group (TMG) as key issues arose. Minor grammatical error3) Would be good to clarify when the Phase 1 and Phase 2 QRI were started as I could not find this in the main text. I recognise the authors tried to highlight this in Figure 1 but this is hampered by the crowded X-axis.4) Were there any factors identified from interviews from patients and recruiting staff which impacted recruitment pertaining to the COVID pandemic? For example, patient fear of making trips to hospitals or clinicians reticence of referring patients to trials during ongoing pandemic.
---

REVIEWER	Prang, Khic-Houy The University of Melbourne, Centre for Health Policy
REVIEW RETURNED	13-Oct-2023

GENERAL COMMENTS	Thank you for the opportunity to read your work and congratulations on reaching your target sample size despite Covid19. This is an evaluation of a QuinteT Recruitment Intervention (QRI) embedded within a randomised controlled trial
--

	(RCT) to improve participants' recruitment. This is a well-written paper and additional information about MARS 2, QRI and UK healthcare system would be helpful for the readers. Specific comments are below:  1. What is the patient journey and referral pathways across the UK healthcare system? When and where are eligible participants invited to participate in the RCT? A figure depicting the patient journey with the RCT process might be useful for the readers given that one of the key barriers is the complex pathway. 2. How many sites and clinicians were involved in the RCT in the UK? Could you provide more information about the roles and responsibilities of each staff in the RCT (e.g., oncologists, surgeons, physicians, nurses, TMG member etc.). It appeared that both nurses and clinicians were responsible for recruitment? 3. The findings suggested that clinical equipoise was a key barrier to recruitment. Could one of the barriers to recruitment also be related to the prevalence of malignant pleural mesothelioma given that it is a rare cancer? How many participants were eligible to participate in the RCT across all sites? Do you have information on how many eligible participants were invited by their clinicians to participate in the RCT but declined and those who were eligible but were never invited by their clinicians? 4. How many immunotherapies pharma sponsored RCTs were rolled out and actively recruiting for the same pool of participants during the same period? Do you have information on how many eligible participants opted to participate in these competing RCTs? Did the sites or clinicians receive financial incentives to recruit participants in the RCT? In pharma sponsored RCTs, sites have quotas and received financial compensation for each participant recruited. 5. There appeared to be variability in how clinicians recruited participants (e.g., clinical judgment and accounting for patient's demographics and circumstances) and the type of treatment information provided. Were clinicians provided a script to follow for participant's recruitment? How many consultations did each clinician conducted throughout the RCT? What feedback did the clinicians received, and how was clinical equipoise specifically addressed? 6. Approximately 30 strategies were selected from the QRI and implemented throughout the RCT. Did all the sites received the same strategies? Which strategies were the most successful in recruiting participants? 7. 25 interviews from 24 participants? Was 1 person interviewed twice? 8. The qualitative findings focused on barriers to participant's recruitment. Did the interviews also explore the impact of the strategies? How acceptable were the strategies to the clinicians? Did the clinicians adopted and implemented any of the strategies provided? Table 2 showed that there were 'monthly recruitment tips in newsletter' and 'recruitment tips document'. Could you provide more information on what the recruitment tips were which may be useful for other cancer RCTs looking to increase recruitment. 9. Participant's recruitment took five years. How does the recruitment rate compare to other rare and non-rare cancer RCTs?
--	--

VERSION 1 – AUTHOR RESPONSE

Reviewer: 1

Dr. Kay Por Yip, University of Birmingham

Comments to the Author:

The authors describe the challenges uncovered in a mixed method approach to investigate recruitment obstacles in the MARS 2 randomised controlled trial. The manuscript highlights how QRI has improved recruitment rate in the trial that it reaches target recruitment numbers. It also describes nicely how unconscious biases and underlying views of clinicians and trial staff can adversely impact recruitment, particularly in the role of cancer trials. I recommend that this manuscript should be accepted. Below are a small list of recommendations proposed

1) First point in the 'Strengths and Limitations' of the abstract will need to be reworded as it is difficult to follow

Response: Thank you for drawing our attention to this. We have now amended the first point to: "A core strength of the study was the ability to analyse actual practice (audio-recording of recruitment discussions), as opposed to relying on reported practice, and to triangulate findings from multiple data sources to gain an in-depth and rich understanding of the key (and often hidden emotive [21,22]) difficulties. This was done in a relatively short time to enable strategies to be devised and implemented to support recruiting staff whilst recruitment was underway. The MARS 2 study took five years to randomise 335 patients. This is a threefold increase in the rate of recruitment compared with the next largest surgical trial for mesothelioma – MARS – which randomised 50 patients in three years [28]."

2) Page 6 line 34 - Findings from Phase 1 were and fed back to the Chief Investigator (CI) and trial management group (TMG) as key issues arose. Minor grammatical error

Response: Thank you, we have since dropped the first 'and'. The sentence now reads: "QRI Phase 1 commenced in March 2018. Findings were fed back to the Chief Investigator (CI) and trial management group (TMG) as key issues arose".

3) Would be good to clarify when the Phase 1 and Phase 2 QRI were started as I could not find this in the main text. I recognise the authors tried to highlight this in Figure 1 but this is hampered by the crowded X-axis.

Response: We have now added dates with regards to the start of QRI Phase 1 and 2 at the end of the Methods section. The text now reads:

"QRI Phase 1 commenced March 2018. Findings were fed back to the Chief Investigator (CI) and trial management group (TMG) as key issues arose. Effective strategies, tailored to address identified issues, were devised and implemented from October 2018 (start of QRI Phase 2). Phase 1 and 2 continued cyclically until recruitment target was reached. QRI training, based on the QuinteT RCT recruitment training intervention [19], was additionally delivered prior to QRI Phase 1 (November 2017 to February 2018) to tackle barriers that had emerged in the pilot phase [20] and previous QRI studies [21-25]."

4) Were there any factors identified from interviews from patients and recruiting staff which impacted recruitment pertaining to the COVID pandemic? For example, patient fear of making trips to hospitals or clinicians reticence of referring patients to trials during ongoing pandemic.

Response: Thank you for raising this point. We had reached sufficient data saturation from the patient and recruiting staff interviews just prior to the start of the COVID pandemic so we had not intended to undertake further interviews. It would have been interesting to focus on this aspect. Unfortunately, the site teams were under increased pressure and had varying capacity to prioritise the study during the pandemic so requesting interviews would have been inappropriate. Instead, we focused on positive encouragement and motivation to help raise the profile of MARS 2 in the final few months of recruitment during the pandemic. Given that you raised this point we have now added a sentence about QRI activity once the study pause for the pandemic had been lifted under the 'Addressing identified recruitment challenges' subheading of the Results section:

"After the COVID-19 study pause was lifted, actions focused on positive encouragement and motivation to help raise the profile of MARS 2 for the final stages of recruitment, respecting the pandemic related pressures that sites were under and the varying capacity that they subsequently had".

Reviewer: 2

Dr. Khic-Houy Prang, The University of Melbourne

Comments to the Author:

Thank you for the opportunity to read your work and congratulations on reaching your target sample size despite Covid19. This is an evaluation of a QuinteT Recruitment Intervention (QRI) embedded within a randomised controlled trial (RCT) to improve participants' recruitment. This is a well-written paper and additional information about MARS 2, QRI and UK healthcare system would be helpful for the readers. Specific comments are below:

1. What is the patient journey and referral pathways across the UK healthcare system? When and where are eligible participants invited to participate in the RCT? A figure depicting the patient journey with the RCT process might be useful for the readers given that one of the key barriers is the complex pathway.

Response: Thank you for this useful suggestion. We have since included a new figure on the study recruitment pathway and highlighted usual clinical practice within this (Figure 1). This has been referred to in the methods section under 'The MARS 2 study' and 'Organisational challenges of conducting mesothelioma research'.

2. How many sites and clinicians were involved in the RCT in the UK? Could you provide more information about the roles and responsibilities of each staff in the RCT (e.g., oncologists, surgeons, physicians, nurses, TMG member etc.). It appeared that both nurses and clinicians were responsible for recruitment?

Response: The new Figure 1 offers more detail on the roles of study site staff. Recruitment was a process rather than a single point in time, and both clinicians and research nurses were involved in speaking with patients about the study at various points, as depicted in Figure 1. It is difficult to calculate how many clinicians were involved in total as this varied over the period of recruitment, but we have now included the total number of sites involved. We have added the following text in the first paragraph of the Methods sections under 'The MARS 2 study' heading:

"Figure 1 summarises the typical study recruitment pathway in the context of usual clinical practice. Adults with a diagnosis of malignant pleural mesothelioma were mostly introduced to the study by respiratory physicians and/or oncologists alongside local research staff at one of 25 medical sites across the UK. Patients were then referred to a thoracic surgeon at one of five trial accredited UK surgical sites (often in different hospitals to the medical site) to determine eligibility and discuss the study in more detail, before typically being referred back to the local medical team for study consent."

3. The findings suggested that clinical equipoise was a key barrier to recruitment. Could one of the barriers to recruitment also be related to the prevalence of malignant pleural mesothelioma given that it is a rare cancer? How many participants were eligible to participate in the RCT across all sites? Do you have information on how many eligible participants were invited by their clinicians to participate in the RCT but declined and those who were eligible but were never invited by their clinicians?

Response: We have now provided the following text in the MARS 2 study section under Methods: "Flow of participants to the point of randomisation - 1030 patients were assessed for eligibility, 645 were eligible and 335 were randomised. Of the 310 eligible patients who did not consent, most were not approached for consent (n=183), did not consent (n=74) or did not undergo two cycles of chemotherapy and repeat CT scan (n=36)."

We have also discussed recruitment challenges in trials of rare diseases – see the following text added to the Discussion section:

"MARS 2 recruitment was further challenged by the condition under investigation being a rare cancer. The rarity of the condition makes finding and recruiting enough patients difficult. Rare disease trials tend to be smaller, longer, and more prone to being terminated, withdrawn or suspended compared with those for non-rare conditions [35]. Missed opportunities for referring and approaching patients, as identified in the present study, become even more impactful when the pool of potentially eligible patients is limited at outset. To add to this challenge, the MARS 2 study was investigating a condition that was very active in terms of research to advance medical treatment opportunities. We know from clinician interviews and screening log data that a very small number of patients being considered for MARS 2 were lost to other studies, but we don't know how many patients were lost before MARS 2 was even considered. This puts a further strain on recruiting to a trial that investigates a rare and life limiting cancer where patients are already difficult to find and where there may be discomfort in approaching them about research at such a time. The value of employing methods to understand and address recruitment challenges in such trials, as with the QRI in the present study, is heightened, as are widespread campaigns to encourage patients to proactively seek involvement in clinical studies they might be eligible for [36]".

4. How many immunotherapies pharma sponsored RCTs were rolled out and actively recruiting for the same pool of participants during the same period? Do you have information on how many eligible participants opted to participate in these competing RCTs? Did the sites or clinicians receive financial incentives to recruit participants in the RCT? In pharma sponsored RCTs, sites have quotas and received financial compensation for each participant recruited.

Response: Thank you for these thought-provoking points but we do not have the information to be able to address them. We do know from screening log data that a very small number of patients (n=10) were ineligible due to co-existing enrolment in another interventional study aiming to improve survival, and at interview a few clinicians told us that they will occasionally decide whether MARS 2 or another trial is more appropriate for a patient. We suspect there was more of this that we did not capture. We have touched upon this in the new text inserted in the Discussion and detailed in the response to your Question 3.

5. There appeared to be variability in how clinicians recruited participants (e.g., clinical judgment and accounting for patient's demographics and circumstances) and the type of treatment information provided. Were clinicians provided a script to follow for participant's recruitment? How many consultations did each clinician conducted throughout the RCT? What feedback did the clinicians received, and how was clinical equipoise specifically addressed?

Response: These are good points. It is not possible to reliably capture how many consultations each clinician conducted throughout the RCT. However, we have now included the following text in the

Results section under 'Addressing identified recruitment challenges' subheading and included the recruitment tips document as a new supplementary material (Supplement 1):

“Support focused on maximising the pool of eligible patients in terms of finding, assessing and approaching them by offering practical solutions, for example sharing site-initiated ideas such as viewing and flagging potentially eligible patients from MDT lists in advance and regular contact with lung nurses to ensure patients were not missed from being assessed for eligibility. Or by raising awareness using evidence from QRI data and opening discussion of how biases and discomforts affected whether patients were considered eligible for the study and approached about it. Awareness was also raised of patients being primed towards or against treatment from colleagues prior to study consideration to indicate the value of recruiters exploring what patients had already been told about treatment options, and to promote sharing of information on MARS 2 with wider colleagues especially those encountered by the patient earlier in the pathway.

Support also focused on ensuring that study discussions offered full, clear, and balanced information. There was a particular emphasis on issues around equipoise and good practice for conveying it. In both individual and group feedback sessions, QRI anonymised data were presented to raise awareness of how imbalances of treatment descriptions, for example disproportionate emphasis on potential risks and benefits or language that subtly favoured or deterred a treatment, could steer patients away from the study and towards or against a particular treatment. Examples from consultation recordings were shared and discussed of more neutral ways of describing treatments and of engaging with patient treatment preferences. The recruitment tips document (Supplement 1) offered suggestions for structuring and discussing recruitment consultations in a way that emphasised the uncertainties around the best treatment, with specific aspects of this spotlighted in monthly newsletters using extracts of QRI data. Feedback and support were set within the recognition of site teams continued above target achievement and encouragement to keep on track.

After the COVID-19 study pause was lifted, actions focused on positive encouragement and motivation to help raise the profile of MARS 2 for the final stages of recruitment, respecting the pandemic related pressures that sites were under and the varying capacity that they subsequently had.”

6. Approximately 30 strategies were selected from the QRI and implemented throughout the RCT. Did all the sites receive the same strategies? Which strategies were the most successful in recruiting participants?

Response: These are useful questions to raise. We have offered information in Table 2 that indicates what actions were performed and who they were delivered to. Any individual feedback was tailored to the individual's consultation and/or interview data. One limitation of the study was that we could not always be certain who saw the information that was shared. As an observational study, with QRI-informed actions implemented throughout a lot of the main recruitment period, it was also difficult to determine which strategies were most successful in recruiting participants. These limitations have been raised in the Discussion section.

7. 25 interviews from 24 participants? Was 1 person interviewed twice?

Response: Yes, one person from a top recruiting site was interviewed a second time over 12 months later to explore why their numbers randomised had declined. We have not specifically stated this level of detail in the paper, but we have made readers aware at the end of the methods section that phase 1 and phase 2 continued cyclically.

8. The qualitative findings focused on barriers to participant's recruitment. Did the interviews also explore the impact of the strategies? How acceptable were the strategies to the clinicians? Did the

clinicians adopted and implemented any of the strategies provided? Table 2 showed that there were 'monthly recruitment tips in newsletter' and 'recruitment tips document'. Could you provide more information on what the recruitment tips were which may be useful for other cancer RCTs looking to increase recruitment.

Response: Most interviews were conducted in the early stage of recruitment in the main trial (QRI Phase 1) so the opportunity to explore the impact of the strategies at interview was limited. We did, however, receive responses from several recruiters acknowledging the value of feedback and how it had raised awareness of certain issues and that they would be mindful of this going forward. A before-after tailored feedback showed evidence of advice being implemented (See "Impact of QRI on optimising recruitment and informed consent" subheading in the Results section and Table 3). Your final question asking for more information on the recruitment tips has been addressed in your Question 5 and inclusion of the Recruitment Tips document as new supplementary material (Supplement 1).

9. Participant's recruitment took five years. How does the recruitment rate compare to other rare and non-rare cancer RCTs?

Response: This is a good point but we have not been able to find any meaningful literature to address this as there are too many varying factors that affect rates. We have however compared the recruitment rate in the MARS 2 study with the next largest surgical mesothelioma trial and noted a substantial increase. We have reported this under the strengths section of the Discussion: "The MARS 2 study took five years to randomise 335 patients. This is a threefold increase in the rate of recruitment compared with the next largest surgical trial for mesothelioma – MARS – which randomised 50 patients in three years [28]".